# Generic Substitutions and Therapeutic Interchanges in Hospital Pharmacies: A Qualitative Study from Western Saudi Arabia

**DOI:** 10.3390/healthcare11131893

**Published:** 2023-06-30

**Authors:** Manar Hassan Alsufyani, Manayer Hammad Alghoribi, Thekra Omar Bin Salman, Asma Fayez Alrabie, Ibtihal Saud Alotaibi, Abdullah Mosa Kharbosh, Mona Yaser Alsheikh, Ali Mofleh Alshahrani, Ahmed Ibrahim Fathelrahman

**Affiliations:** 1College of Pharmacy, Taif University, Taif 21944, Saudi Arabia; imanarhs@gmail.com (M.H.A.); manayer.g@hotmail.com (M.H.A.); thekrapharma@gmail.com (T.O.B.S.); afr1996@outlook.sa (A.F.A.); ibtihalsaud@hotmail.com (I.S.A.); 2Department of Clinical Pharmacy, College of Pharmacy, Taif University, Taif 21944, Saudi Arabia; a.kharbosh@tu.edu.sa (A.M.K.); a.mona@tu.edu.sa (M.Y.A.);

**Keywords:** generic substitutions, therapeutic interchanges, hospital pharmacists’ views and practice, clinical pharmacy, pharmaceutical care, rational dispensing

## Abstract

**Background:** The aim of the present study was to understand hospital pharmacists’ views and practices regarding generic substitution and therapeutic interchange. **Method:** This was a qualitative study involving pharmacists from three Western Saudi governorates: Taif, Makkah, and Jeddah. It included respondents from the Ministry of Health (MoH), military and private hospitals. Pharmacists were selected using a convenient sampling technique and data were collected using a structured face-to-face interview. **Results:** Fifty-seven pharmacists agreed to participate in this study. In MoH and private hospitals, generic substitution is a pharmacist-initiated act, while therapeutic interchange requires physician approval. Medication unavailability, side effects, patient characteristics, outcomes, and economic status justified most substitution decisions. In military hospitals, both types of substitutions are controlled by an auto-switch policy and physicians should be informed. In all hospitals, there are policies regulating substitution. Medications eligible for interchange mentioned by pharmacists from different hospitals were comparable to some extent. Pharmacists from the private sector considered substitution a supportive economic measure for both hospitals and patients. Most pharmacists highlighted that patient convenience and physician approval are the most challenging situations in substitution practice. **Conclusions:** An enhanced understanding of substitution and knowledge about medications included in the hospital formulary will be valuable support to the implementation of substitution practice which responds to the patients’ needs to improve their outcomes.

## 1. Introduction

Generic substitution (GS) is a common dispensing practice, defined as replacing a prescribed medicine with another alternative that has the same active drug content from another manufacturer (i.e., a generic drug) [1]. Therapeutic interchange (TI) is the replacement of a prescribed medicine with another one that differs in the active constituents but is thought to be therapeutically equivalent to one in the same drug class (i.e., within-class substitution, such as switching from atorvastatin to simvastatin) or to one from a different class (i.e., between-class substitution, such as switching from an angiotensin-converting enzyme (ACE) inhibitor to an angiotensin receptor blocker (ARB)) [2].

The drugs used for TI might not be similar in terms of pharmacokinetics or chemical characteristics; they may differ in their mechanisms of action, toxicity, adverse reactions, and drug interaction profiles [3]. No consensus has been reached among countries regarding the policies governing TI. However, a common feature is a requirement for a protocol describing which medication can be substituted without risk [1]. Differences exist among countries regarding the requirements regulating generic dispensing. In most situations, not all drugs can be directly substituted just because claims state they are equivalent: a well-defined list of medications suitable for GS is required [1].

Possible motivators for substitutions include a cheaper medication being available, a medication being removed from insurance coverage, a prescriber receiving an incentive to prescribe a certain manufacturer’s products instead of others, or a prescriber switching to a more effective medication to meet patient treatment goals [4].

Therapeutic substitution is affected by some ethical dilemmas, particularly when the switch is within or between classes. To make an informed ethical decision, five questions should be asked. If the answer to any question is “no”, then the switch should not be made [2]: (1) Has the patient provided written informed consent to the switch? (2) Is the overall cost of drugs truly lower after considering all relevant costs rather than focusing simply on the medicine acquisition cost? (3) Is this switch between drugs in the best interest of the patient? (4) Does the alternative product produce a therapeutically equivalent outcome? (5) Do robust data exist on its safety, efficacy, and quality? To ensure the safety of GS or TI and to improve patient adherence, an alternative option should be selected considering factors such as additive substances, labeling, and packaging. To avoid prescribing and/or dispensing errors, prescribers should be encouraged to use the international nonproprietary names of the prescribed medicines [5].

The physician’s decision to replace a patient’s pharmaceutical product with another having the same therapeutic intent is defined as switching drugs [5]. Numerous studies have been conducted worldwide on the issue of generic substitution and therapeutic interchange [6,7,8,9,10,11,12,13,14,15,16,17,18,19,20,21,22,23,24]. However, few studies have been conducted in Saudi Arabia, and these have mostly been quantitative studies [11,20]. Qualitative studies are considered most appropriate for “how?” and “why?’’ questions exploring processes and patterns in people’s thoughts and behavior. Recently, the concern regarding the use of qualitative research has increased among healthcare researchers and, to a lesser extent, among pharmacy practice researchers due to a culture of preference for established quantitative research [25]. However, few examples exist of pharmacy-practice-related qualitative studies conducted worldwide [26,27,28,29,30]. Qualitative research approach provides the opportunity for pharmacy-practice researchers to discover, understand, and explain the opinions and views of healthcare workers, consumers, and patients [31]. To the best of our knowledge, the use of qualitative research in assessing and understanding practice-related issues in Saudi Arabia has been limited [27]. In 2014, Albader and Khan qualitatively studied 20 community pharmacists in Alahsa, Saudi Arabia, to understand the factors affecting pharmacists’ decisions to dispense cheap or expensive medications. They found that pharmacists from hospital-affiliated pharmacies were more concerned about the quality of the medication before dispensing it to a patient, whereas pharmacists from non-hospital-affiliated pharmacies were more supportive of therapeutic substitution, which influenced their decision to dispense cheaper generic or expensive branded medicines [27].

Our aim in this study was to understand hospital pharmacists’ views and practices regarding generic substitution and therapeutic interchange during dispensing. In particular, we aimed to gain an in-depth understanding of the following issues: (1) the use of substitutions and the available types of substitution, (2) which medications are subjected to substitution, (3) the reasons for substitution, (4) normal processes and procedures for substitution, (5) challenges facing substitution practices, (6) the existence of official policies and protocols supporting substitution in hospitals, and (7) pharmacist beliefs in the importance of medication substitutions.

## 2. Materials and Methods

### 2.1. Setting and Study Design

A qualitative study was conducted during February and March 2021, among hospital pharmacists in three populous governorates: Taif, Makkah, and Jeddah. A total of 136 governorates represent the second-level administrative divisions that form the 13 regions in Saudi Arabia. These three governorates are geographically located in the western region of Saudi Arabia and administratively belong to the Makkah region. The region is important as a destination for Muslim pilgrims who travel from all over the world every year to perform Hajj during a specific period (for two months of the Hijri calendar: Dhul-Qidah and Dhul-Hijjah) or to perform Umrah during all the other months of the year. The population of the Makkah region represents about one-quarter of the total Saudi population. The healthcare services, including pharmaceutical care, in this region are well-developed and well-managed, serving both residents and visitors. However, to what extent such services are comparable across different service providers is unknown. Thus, we decided to include a sample representing the three main service providers: the Ministry of Health (MoH) hospitals, private hospitals, and military hospitals.

### 2.2. Study Population and Sample Size

#### 2.2.1. Inclusion Criteria

We targeted the main hospitals in the region where Taif University pharmacy students regularly receive training. All pharmacy staff were the targets for interviews, with a particular focus on representing pharmacy technicians, pharmacists, clinical pharmacists, medication safety officers, pharmacy and therapeutic committee member pharmacists, drug information pharmacists, and pharmacy managers/directors.

#### 2.2.2. Sample Size

The hospitals and the number of pharmacists to be included were determined based on achieving a saturation status. Saturation is considered to be reached in qualitative research when no additional information is being obtained from the interviewees (i.e., additional interviews yield repeated information mentioned by the previously interviewed subjects) [32]. In this study, we looked for saturation within each hospital category (i.e., MoH, military, and private).

### 2.3. Sampling

Due to the absence of a sampling frame, hospital pharmacists were selected using a convenience sampling technique within a group of hospitals from each of these selected cities. Convenience sampling is a nonprobability sampling technique that is commonly used for qualitative research [33]. The interviewers approached all pharmacy staff who were available at the hospitals on the days of the visit. Three pharmacy staff refused to participate in the hospitals that approved the study. All three were from private hospitals. The administration of two private hospitals, one in Jeddah and the other in Makkah, did not approve the study as a prophylactic measure during the COVID-19 lockdown.

### 2.4. Data Collection Tool

Data were collected using structured face-to-face interviews, which aimed to obtain information on seven main themes in addition to the sex, job title, academic qualification, number of experience years, and practice setting of the respondent. The main themes that were discussed with the interviewee were decided in advance based on a general reading of the literature and on our own experience about the topic, which was described by Bush and Amechi as a deductive coding approach [33]. Data were collected by Pharm-D interns using audio recordings. The interns were taught about all aspects of qualitative research as part of the preparation to conduct their graduation project.

All interviews were recorded, and the transcripts were confirmed by the participants. Prompting and probing were used during the interviews, when needed, to develop a more thorough understanding of the respondents’ feedback. The closing at the end of each interview contained a key component, and the average interview duration ranged from 30 to 60 min.

### 2.5. Validation of Interview Plan

The preliminary plan for the interview was tested with a member of faculty from the College of Pharmacy, Taif University, who had extensive experience in hospital settings. They worked as a clinical pharmacist in various wards, held different positions, and were the head of a pharmacy department before joining academia. We interviewed them for 1.5 h. At the end of the interview, they were asked to provide feedback on the interview outline and to ask questions; they provided constructive suggestions and recommendations for edits.

### 2.6. Thematic Analysis

Our thematic analysis consisted of seven themes: (1) the use of substitutions and available types of substitution, (2) which medications are substituted, (3) reasons for substitution, (4) normal processes and procedures for substitution, (5) challenges facing substitution practices, (6) existence of official policies and protocols supporting substitution in hospitals, and (7) pharmacist beliefs in the importance of substitutions. Five researchers manually transcribed the interviews, and each one checked the work of the others to confirm the consistency of the recoding.

### 2.7. Ethical Considerations

Ethical approval was obtained from the Taif University Ethics Review Board. Approval from the MoH, military hospitals, and private hospitals was obtained prior to conducting the interviews. Pharmacists who agreed to participate within the hospitals that had approved the study were interviewed after providing written informed consent.

## 3. Results

### 3.1. Characteristics of Interviewed Pharmacists

Fifty-seven pharmacy staff agreed to participate in the study and were interviewed. Twenty-one, fifteen, and twenty-one pharmacists were interviewed from MoH, military, and private hospitals, respectively. More details on the participant characteristics are shown in Table 1.

### 3.2. Theme 1: Presence of Substitutions and Available Types of Substitution

All the interviewed pharmacists from different regions and hospital types declared the use of generic substitution (GS) and therapeutic interchange (TI), with few exceptions. One pharmacist from a private hospital and six pharmacists from the MoH declared the use of generic substitution only. However, such respondents had relatively less experience. Four of them were pharmacy technicians, one was a clinical pharmacist with only two years of experience, and one was a holder of a bachelor of pharmacy degree with five years of experience. All other respondents employed as clinical pharmacists, medication managers, or safety officers agreed that both GS and TI were used.

### 3.3. Theme 2: Which Medications Are Subjected to Substitution?

The most common medications subjected to GS in MoH hospitals were paracetamol, levetiracetam (Keppra^®^), atorvastatin, clopidogrel, quetiapine, and olanzapine; for TI, the medications were antibiotics, antipsychotics, anticoagulants, antihypertensives, statins, and antidepressants (Table 2).

For military hospitals, the pharmacists mentioned gliclazide, perindopril, amlodipine, omeprazole, levetiracetam (Keppra^®^), insulin, valsartan, nitrofurantoin, and ASA as examples of GS medications (Table 2). Regarding TI, PPIs, NSAIDs, antihypertensives, and statins were mentioned as the most common classes of GS drugs.

Pharmacists in private hospitals declared that amlodipine, amoxicillin, atorvastatin, and clopidogrel were examples of GS drugs, whereas statins, beta-blockers, PPIs, and CCBs were mentioned as examples of TI (Table 2).

### 3.4. Theme 3: Reasons for Substitution

Unavailability, side effects (SEs), patient characteristics, patient outcomes, and economic status were the main reasons pharmacists decided upon drug substitution. We noted agreement on this finding among pharmacists from all three categories of hospitals.

### 3.5. Theme 4: Normal Processes and Procedures for Substitution

When asked about the procedures and processes regarding substitution, respondents from MoH and private hospitals declared that a generic substitution is a pharmacist-initiated act, whereas therapeutic interchange must be approved by a physician, with few exceptions. For example, a pharmacist from the MoH said “We do generic substitutions and therapeutic interchange automatically for drugs in the medications list included in a protocol approved from the MoH, but the pharmacist needs physician approval for any drug not on the list”. One pharmacist from the MoH mentioned another solution for out-of-stock medications instead of substitution: “if a medication is not available at the hospital, I would source it from other hospitals via departments of medical supply”. Two pharmacists from private hospitals mentioned patient approval as an additional requirement for substitution. A pharmacist from the MoH said: “Therapeutic interchange is conducted after consulting the physician as it’s not based on the pharmacist only. Quality pharmacists send clarification forms that contain the pharmacists’ suggestion for therapeutic interchange, and after that, the physician changes the prescription, while generic substitution is a pharmacist-initiated act”.

According to military hospital pharmacists, both generic substitution and therapeutic interchange do not require prescriber approval as long as the substituted medications are included in the automatic switching list; otherwise, physician approval is required. However, in all cases, the prescriber is informed about the substitution. If a substitution is needed for an inpatient order, it is performed by clinical pharmacists using the electronic hospital system, so the prescriber is aware of the substitution via the electronic system (i.e., progress notes). For outpatients, pharmacists contact the prescribers via telephone to inform them about the substitution. However, a pharmacist from a military hospital said that “To do therapeutic interchange, we must call the physician to accept the drug change”.

### 3.6. Themes 5: Challenges Facing Substitution Practices

When asked about the challenges facing them during drug-switching processes, respondents from the MoH stated the following: the unavailability of alternative medications, patient refusal to accept the new medication, poor communication with physicians, difficulty convincing a patient or a physician to switch, and lack of better alternatives.

Pharmacists from private hospitals mentioned challenges like the difficulty of convincing patients to accept the different drugs, notably if they were comfortable with their medications, especially chronic diseases patients; physicians’ hesitancy to agree to substitution or difficulty in contacting physicians; less experience with and knowledge regarding substitution. One pharmacist from a private hospital mentioned, “I did not face any challenges during substitution practices because of good communication with prescribers”. Another pharmacist from a private hospital said, “thirty percent of patients agree to substitution, sixty percent ask to call the physician first, and ten percent do not agree”.

Pharmacists from military hospitals mentioned challenges such as patient acceptance and preference for their usual medications, the lack of alternative drugs, the lower efficacy of the substituted medication, and the drug being restricted for prescription by a specialist only. Another challenge noted by pharmacists with less experience was being unauthorized to perform substitutions. Additionally, two pharmacists stated that they would face no challenges if a clear policy and guidance was in place for everyone. Pharmacy managers at military hospitals reported that they resolve many challenges facing substitution using the Wasfaty website, which provides the nearest pharmacy with the drug in case it is unavailable at the current location.

### 3.7. Theme 6: Official Policies and Protocols Supporting Substitution in Hospitals

When we asked pharmacists from MoH hospitals about the protocols for substitution, most of them (*n* = 17) answered, “yes there is a protocol approved by the MoH”, whereas others (*n* = 4) answered “no”. Some pharmacists said that the substituted medications list is updated when needed depending on the medication supplied. One pharmacist was not aware of any specific policy, “we just write that the medication is unavailable, and we suggest a medication for substitution”.

#### 3.7.1. Substitution Policy of MoH

The policy from the MoH for substitution states:The pharmacist can automatically substitute a medication on the formulary list approved by the MoH.Automatic substitution is only allowed after exhausting all efforts to secure the stock of the prescribed medication according to the out-of-stock policy.Pharmacists apply this policy to patients under the care of the MoH only.

#### 3.7.2. Substitution Policy of Private Hospitals

The pharmacists from the private hospital said they have an internal policy regarding substitution. For example, the policy from one private hospital requires the following:Enough information about the patient should be available for generic substitution to be performed.Therapeutic interchange should be applied when the same generic medications are unavailable after being approved by the physician. If not approved, the patient should be transferred to another hospital.

When asked, a pharmacist from a private hospital said “we have a formulary system policy for alternative medication”. The pharmacist said that they also use the MoH policy, CBAHI Standard, Saudi FDA, and manufacturer protocols. Many pharmacists from private hospitals said that substitution depends on the pharmacy and therapeutics committee and the clinical pharmacists who are responsible within each medical institution.

#### 3.7.3. Substitution Policy of Military Hospitals

The pharmacists from military hospitals also have a protocol for substitution called the auto-switch policy, which is based on a pharmacy and therapeutic committee. The policy identifies a group of drugs that can be substituted with certain agents assumed to be therapeutically equivalent as well as some requirements that must be filled prior to substitution.

### 3.8. Theme 7: Pharmacist Beliefs Regarding Medication Substitutions

Pharmacists from all types of hospitals believe that substitution is important for patients and that pharmacists must have proper knowledge and skills to perform drug substitutions.

Most of the pharmacists in the military and MoH hospitals conduct generic substitution and therapeutic interchange after informing the physician or by following a protocol supporting this process.

Most pharmacists at private hospitals conduct substitutions to support the financial economy of the hospital. One pharmacist stated that substitution is beneficial for the patient’s economic status. A pharmacist said, “substitution is necessary to save patients’ lives and avoid condition-worsening when a drug is unavailable. If a company is bankrupt or a drug manufacturer is no longer available, a substitute drug is available and that saves time for the patient”.

## 4. Discussion

Our findings revealed that both GS and TI are conducted in hospitals in western Saudi Arabia. The policy makers in the Ministry of Health and military hospitals as well as some private hospitals considered identifying certain classes for substitution and interchange, which is called automatic drug-switching. Ideally, medications selected for the auto-switch list are expected to be therapeutically comparable in terms of effectiveness and safety and available at variable prices to reduce costs via substitution. Johansen and Richardson conducted a cross-sectional study involving an economic component regarding savings via therapeutic substitution. They discovered a pattern of increased expenditure among 26 drug classes associated with favoring branded drugs over generic ones when the substitution was possible [12]. Increased expenditure was highest with statins, atypical antipsychotics, proton pump inhibitors, selective serotonin reuptake inhibitors, and angiotensin receptor blockers. Notably, most of the medications included in this list were mentioned by the pharmacists who were interviewed in our study.

In this study, a variety of medications were subjected to substitution in all MoH, military, and private hospitals. However, this practice is controlled by regulating policies. Globally, both generic substitution and therapeutic interchange are acceptable. Even some medications considered to be dangerous are included elsewhere in lists of medications available for substitution, such as narrow therapeutic index medications, those that are biosimilar, and narcotics. In Poland, Pawlowska et al. surveyed hospital pharmacists to explore their views regarding the substitution of biosimilar medicines [7]. Pharmacists reported supportive views. In Australia, community-based pharmacists were willing to provide opioid substitution treatment to their patients [24]. In addition, most of the pharmacists provided this service. In the U.S., most pharmacists were supportive of substituting narrow therapeutic index (NTI) drugs, and the majority considered the practice to be effective and safe. In terms of practice, most of them performed generic NTI substitution for prescriptions [8].

Unavailability, side effects (SEs), patient characteristics, patient outcomes, and economic status represented the important reasons pharmacists decided to conduct a substitution. Here, agreement was expressed among the pharmacists from the three categories of hospitals. Nokelainen et al. surveyed 1043 community pharmacists in Finland to determine the reasons for accepting or refusing generic substitution among patients. The findings showed that the leading cause of accepting substitution was a desire to reduce the expenses and the unavailability of the prescribed or preferred medicines. Common reasons for refusing substitution were a nonsignificant cost difference between the prescribed medication and the alternative and patient satisfaction with a previously used medicine. Factors strongly influencing patients’ choice of a substituted medication were cost, familiarity, and availability [6].

According to the current study, in the MoH and private hospitals, generic substitution is a pharmacist-initiated act, whereas therapeutic interchange requires physician approval. In military hospitals, both types of substitution are governed by an auto-switch policy, so physician approval is not needed but the physician must be informed. According to Pawlowska et al., pharmacists involved in the dispensing of biosimilars acknowledged their importance in reducing medication costs, but they do not recommend substitution without prior permission from the prescriber [7]. The authors highlighted the importance of controlling substitutions with biosimilars using specific policies, promoting effective communication between prescribers and dispensers, and initiating educational intervention to promote the safe and effective handling of biosimilars. In Australia, Lawrison et al. assessed community-based pharmacists’ attitudes and practices regarding opioid substitution treatment. They concluded that policies aimed at retaining pharmacists, especially in resource-poor rural regions, could be considered to support a collaborative approach between general practitioners and pharmacists [24]. According to Drozdowsk and Hermanowski, who conducted a study in Poland, letting a patient choose a generic substitute is a patient right [14]. However, only 40% of the surveyed sample of pharmacists always informed patients about that right. In Australia, Chong et al. performed a nationwide study on generic substitution [21]. They concluded that patient acceptance optimizes the generic medicine used.

When asked about the challenges faced during drug-switching processes, the respondents from the MoH mentioned the unavailability of alternative medication, patient refusal to accept different medications, poor communication with physicians and difficulty with convincing physicians, a patient or a physician lack of preference to switch medications, and a lack of better alternatives. Pharmacists at private hospitals mentioned among their challenges the difficulty of convincing patients to accept the substituted drugs if they are comfortable with their medication, especially chronic diseases patients; physicians’ reluctance to accept substitution or difficulty in contacting them and less experience and knowledge regarding substitutions. Pharmacists at military hospitals mentioned among their challenges patient acceptance and preference for their usual medications, the lack of alternative drugs, the lower efficacy of the substituted medication, and the drug being restricted for prescription by specialists only. El-Jardali et al. conducted a mixed quantitative and qualitative study to address community pharmacists’ views and practices regarding a new generic drug substitution policy in Lebanon [10]. The barriers reported by pharmacists included the lack of interest among end users to adopt the substitution policy; poor compliance and the absence of transparency about the goals, responsibilities, and duties of each party. The abuse of substitution options without enough justification was also identified as a major hurdle. This situation has been worsened by a lack of any measure for monitoring and controlling the prescription pattern of physicians. The challenges faced by some pharmacists in terms of convincing prescribers and patients to accept substitution when indicated could be solved by being knowledgeable about the medications, the benefits of substitution, and the policies regulating them. Euen and Fadda mentioned the importance of pharmacists being knowledgeable about FDA therapeutic equivalent standards to understand the criteria and be able successfully to communicate them to patients and healthcare professionals [9].

Our study revealed the existence of official policies and regulations for generic substitution and therapeutic interchange in all types of hospitals. However, pharmacists practicing in hospitals should be made aware of such policies to be able to practice in agreement with the requirements and be confident in implementing the policy. Most of the pharmacists interviewed in this study were found to be aware of a policy, but their awareness of the details was insufficient. In the same context, a study conducted among Indiana, U.S., community pharmacists found a severe lack of awareness about FDA therapeutic equivalence requirements [9]. The authors concluded that both patients and prescribers doubt the safety and efficacy of generic drugs. Pharmacists must have in-depth knowledge about FDA therapeutic equivalence to effectively communicate the relevant and needed information to their patients and healthcare providers. If a policy is not enforced and a system for monitoring the implementation is set in place, substitution policies seldom produce the intended effects. For example, El-Jardali et al. revealed problems in the implementation of a substitution policy adopted in Lebanon [10].

Pharmacists from all types of hospitals think that substitution is important for patients and that pharmacists must have proper knowledge and skills to practice substitution. A study was performed in central Saudi Arabia among community pharmacists to determine their knowledge and perception of generic medicines [34]. Most respondents thought that a generic equivalent is as effective as the initial product. More than two-thirds of the respondents thought that a generic medication must be similar to the brand medicine in both dose and dosage form. Approximately one-half of the participants believed that side effects are more common with generic medicines than with brand-name medicines. The study by El-Jardali et al. in Lebanon indicated strong support among community pharmacists for the generic substitution policy, although less than half of them practiced substitution [10]. A survey conducted among pharmacists in the U.S. explored their beliefs about generic substitution with narrow therapeutic index (NTI) drugs; the majority of respondents considered the practice to be effective and safe. In terms of practice, most of them always performed generic NTI substitution for initial prescriptions, and a substantial proportion of them did so for refills [8]. A qualitative study of 16 Swedish community pharmacists addressed the issue of substitution [18]. The pharmacists stated that GS reduces medication costs. However, they reported a worry about the confusion that switching might cause patients, which would affect their adherence and prevent them from achieving the intended therapeutic outcomes. Participants thought that GS has shifted the attention in pharmacist–patient communications toward regulations and economic aspects. Another study in Ireland interviewed 44 community pharmacists about generic medicines [15]. The study revealed positive attitudes toward these generics and the regulation that supports their use.

In the current study, most of the pharmacists in the military and Ministry of Health hospitals accepted generic substitution and therapeutic interchange after informing the physician or following the protocol supporting this process. Most pharmacists in private hospitals conducted substitution to support the financial economy of the hospital. One pharmacist considered substitution beneficial for the patient’s finances. In Finland, Nokelainen et al. demonstrated that a medicine’s price was the most influential factor in the patient’s decision to accept or refuse GS and in the selection of the option to substitute the prescribed medication [6]. This was consistent with the findings of another study from Poland by Pawlowska et al., who surveyed hospital pharmacists to explore their views about substitution with biosimilar medicines [7]. Most respondents stated that the main advantage of biosimilars is related to decreased costs. Johansen and Richardson estimated the possible savings through therapeutic substitution, and they concluded that substitution offers a potential mechanism to substantially decrease drug costs if the implementation does not adversely affect the quality of care [12].

The current study addressed a vital pharmaceutical-care and health-policy-related issue and revealed some important findings. Therapeutic substitution requires the continuous attention of researchers, practitioners, and policymakers, as it represents a recent global concern. Therapeutic substitution has been proposed as one of four strategies in interventional pharmacy economics in cost-constrained environments [35] and as a strategy for expanding the role of pharmacists during the era of pandemics because it would help with reducing medication shortages and ensuring the sustainability of supply chains [36,37,38].

## 5. Study Limitations

A limitation of qualitative studies is that the information collected from respondents might be affected by bias as they may report what they think or perceive rather than what is happening. However, in the current study, we included pharmacy staff from different backgrounds including staff working in a variety of roles, with a range of experience, and from three different hospital types; additionally, the information provided by most of them was consistent, with few exceptions. In addition, the findings of qualitative studies are not intended to be generalized. These studies are conducted to understand a certain aspect within a particular setting, so the situation may differ in other regions and other practice settings.

## 6. Conclusions and Recommendations

Our findings revealed the application of both generic substitution and therapeutic interchange in the studied hospitals including MoH, military, and private sector hospitals. Policies are in place for substitution, with a specific list of medications eligible for substitution. The current study provides baseline information; the study can be replicated in other regions to confirm the findings. An enhanced understanding of substitution and knowledge about medications included in the hospital formulary can provide valuable support for the implementation of substitution practices, which respond to patient needs to improve their outcomes.

## Figures and Tables

**Table 1 healthcare-11-01893-t001:** The distribution of pharmacy staff participants by demographics.

Characteristic	n (%)
**Governorate**	
Taif	39 (68.4%)
Jeddah	14 (24.6%)
Makkah	4 (7%)
**Type of hospital**	
MoH	21 (36.8%)
Private	21 (36.8%)
Military	15 (26.4%)
**Sex**	
Male	30 (52.6%)
Female	27 (47.4%)
**Highest qualification**	
PharmD.	26 (45.6%)
B.S. Pharm	21 (36.8%)
Diploma	5 (8.8%)
Master	5 (8.8%)
**Job title**	
Pharmacist	23 (40.4%)
Clinical pharmacist	15 (26.3%)
Pharmacy manager	10 (17.5%)
Technician	5 (8.8%)
Quality manager	3 (5.3%)
Medication safety officer	1 (1.7%)
**Years of experience**	
>5	36 (63.2%)
≤5	21 (36.8%)

**Table 2 healthcare-11-01893-t002:** Medications subjected to substitution.

	MoH	Military	Private
**Generic Substitution**	Paracetamol	Gliclazide	Amlodipine
	Levetiracetam (Keppra ^®^)	Perindopril	Amoxicillin
	Atorvastatin	Amlodipine	Atorvastatin
	Clopidogrel	Omeprazole	Clopidogrel
	Quetiapine	Levetiracetam (Keppra ^®^)	Diclofenac
	Olanzapine	Insulin	Captopril
		Valsartan	Perindopril
		Nitrofurantoin	
		ASA	
**Therapeutic Interchange**	Antibiotic	PPIs	Statins
	Antipsychotic	NSAIDs	Beta blockers
	Anticoagulants	Antihypertensives	CCBs
	Antihypertensives	Statins	Antihistamines
	Statins		
	Antidepressants		

## Data Availability

The raw data supporting the conclusions of this article will be made available by the authors, if needed.

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
