# Peer review of "Generic Substitutions and Therapeutic Interchanges in Hospital Pharmacies: A Qualitative Study from Western Saudi Arabia"

_healthcare, 2023, doi:10.3390/healthcare11131893_

Round 1
Reviewer 1 Report
Reviewers' comments
Medicine substitution is integral to a pharmacist’s role in ensuring cost saving on pharmaceuticals and that patients get the best possible care amid the existential challenges in service delivery. The process can be simple or complex depending on the level of care, the disease’s condition, availability issues, ethical cost, desired patient outcome constraints, knowledge of the pharmacist and so many other factors. In this article, the authors outline using a structured interview the intricate process of medicines substitution in some hospitals in the Saudi Arabian region. These results are timely, in informing pharmacy practice locally and globally.
Below are a few comments to improve the overall work done by the authors.
The keywords
The following keywords can be added; Clinical Pharmacy, Pharmaceutical care, rational dispensing
Introduction
Line 24 – 45: The authors might want to revise those statements
The method
Comment: Line 120 – 122: in this section, the authors presented the study design and settings. Whiles the design is fine I think more information is needed for readers to understand the setting well.
v Why did the authors choose these settings?
v What are governorates and how many of them are there in the country?
v What is the distribution of hospitals (Public and private) in these regions?
v Brief geography of the selected regions.
v Perhaps population proportions and economic status might also be essential.
Comment: In lines 125 – 128 the authors presented the inclusion criteria of study participants, whiles this approach is fine it does not give a clear picture. I will strongly recommend that the sampling be presented in two clusters, firstly the regions are selected based on predefined criteria then the hospital and the participants.
Comment: Who are medication safety officers, are they also pharmacists?
Comment: In lines 129 – 132 the authors stated that the participants were sampled to saturation. Can you give more details about this process? How did you conclude that you have reached saturation in this specific study?
Comment: Give more detail on the sampling process stated in lines 134 – 135. How were participants approached, did all those approached consent and participated in the study?
Comment: In line 136 the authors talk about the data collection tool, but nothing was stated about how it was made, what informed its contents, and if it was validated or not. Kindly add that.
Comments: There is nothing stated about the data collectors, where they trained, did they take part in the pilot study. How certain are you that they were all competent enough to collect information that is aligned with the study objectives? I also realized that the tool was not piloted on real participants but on ” a faculty staff from the college of Pharmacy, Taif University who has a long experience in a hospital setting ”. is this person a pharmacist, he/she previously engaged in the activity of GS or TI, if this is not so, why was he/she selected? For sure a faculty staff might be competent enough to advise on the process of the interview but I am not sure if they can on the type of questions to ask about GS or TI.
Comment: How was the data managed? Who did the transcribing, was it done using software or manually? Was the transcripts checked and rechecked for consistency with the recording?
The results
Comments: Line 168. There is nothing about the characteristics of the participants, here you just mentioned the number of participants from each of the hospitals.
Comments: Line 176: in the results section there is no need for justifying anything only present the results. Do justification in the discussion section.
Comments: Line 200 – 224, In the private setting, I did not see anything about the process of GS and TI. From the interview what was the response from pharmacists in the community to that specific question?
Comments: line 250 “Those who said “no” are having low experience in MoH hospitals and have 250 no background of the available protocol “How did you know they have low experience in MOH protocols?
Discussion
Comments: There are many limitations in this study the authors did not account for there should be a sub-section in the discussion that elaborates on those limitations.
Conclusions
Comments: There is so much information generated yes the conclusion is redundant. Perhaps the authors might want to provide insight and recommendations for policy briefs.
I think the English Language needs to be revised by an expert. There are some grammatical errors.
Author Response
Reviewer 1
Reviewers' comments
Medicine substitution is integral to a pharmacist’s role in ensuring cost saving on pharmaceuticals and that patients get the best possible care amid the existential challenges in service delivery. The process can be simple or complex depending on the level of care, the disease’s condition, availability issues, ethical cost, desired patient outcome constraints, knowledge of the pharmacist and so many other factors. In this article, the authors outline using a structured interview the intricate process of medicines substitution in some hospitals in the Saudi Arabian region. These results are timely, in informing pharmacy practice locally and globally.
Below are a few comments to improve the overall work done by the authors.
Authors responses: The authors would like to thank the reviewer for his constructive comments and critics that have contributed much to the improvement of the manuscript.
The keywords
The following keywords can be added; Clinical Pharmacy, Pharmaceutical care, rational dispensing
Authors responses: Done, the suggested keywords are added now.
Introduction
Line 24 – 45: The authors might want to revise those statements
Authors responses: Done, this section has been edited according to the reviewer suggestion.
The method
Comment: Line 120 – 122: in this section, the authors presented the study design and settings. Whiles the design is fine I think more information is needed for readers to understand the setting well.
v Why did the authors choose these settings?
Authors responses: Done, the justification has been provided under setting and study design.
v What are governorates and how many of them are there in the country?
Authors responses: Done, an explanation has been provided under setting and study design.
v What is the distribution of hospitals (Public and private) in these regions?
Authors responses: According 2021 data from Saudi M.o.H. that is based on number of hospital beds across Saudi Arabia, the coverage of Ministry of health hospitals, other public hospitals including military ones, and the private sector are 58.7%, 18.1%, and 23.2%, respectively (Statistical YearBook. A report from Saudi M.o.H. 2021. Page 56). According to a report on public-private dynamics in Saudi healthcare provision backdated to 2018, there was 14public hospitals and 31 private hospitals in Makkah region (Alhowaish, A., & Alshihri, F. (2018). Regional inequalities and public-private dynamics in Saudi healthcare provision. Global Science Journal, 6(1), 34–43).
The hospital covered by this study were the big main hospitals in the region where our pharmacy students receive their introductory and advanced clinical training.
Since our study was qualitative, the authors were not concerned about representing all hospitals in the region numerically (i.e., in qualitative research certain sample size is calculated to represent the numbers of hospitals and number of pharmacy staff). However, qualitative research is intended to describe how substitution practices is practiced in the region and the required sample is achieved once the interviews yield no additional information. In addition, the responses provided by respondents were consistent across different types of hospitals in the region. We provided additional reference to explain saturation (number 33) which has been discussed also in other various resources.
v Brief geography of the selected regions.
Authors responses: Done, a brief explanation has been provided under setting and study design.
v Perhaps population proportions and economic status might also be essential.
Authors responses: Done, a brief explanation has been provided under setting and study design
Comment: In lines 125 – 128 the authors presented the inclusion criteria of study participants, whiles this approach is fine it does not give a clear picture. I will strongly recommend that the sampling be presented in two clusters, firstly the regions are selected based on predefined criteria then the hospital and the participants.
Authors responses: in qualitative studies, convenience sampling technique is the most used sampling approach. Multistage clustering sampling is a probability sampling technique commonly used for quantitative research because the first concern in quantitative studies is representativeness regarding the varieties of subjects within population and regarding population size. We have added a brief description of convenience sampling technique under inclusion criteria and sampling subtitles. As mentioned above we targeted the main big hospitals in the region where our students are trained regularly, and recruitment was stopped once saturation was achieved.
Comment: Who are medication safety officers, are they also pharmacists?
Authors responses: Yes, all respondents are pharmacy staff the same like drug information pharmacists and the others. This categorization was based on the nature of the jobs they involve in.
Comment: In lines 129 – 132 the authors stated that the participants were sampled to saturation. Can you give more details about this process? How did you conclude that you have reached saturation in this specific study?
Authors responses: Done, a brief explanation has been provided under Sample Size. A reference is added to support the statement (reference number 33).
Comment: Give more detail on the sampling process stated in lines 134 – 135. How were participants approached, did all those approached consent and participated in the study?
Authors responses: Done, a brief explanation has been provided under sampling. A reference is added to support the statement (reference number 34).
Comment: In line 136 the authors talk about the data collection tool, but nothing was stated about how it was made, what informed its contents, and if it was validated or not. Kindly add that.
Authors responses: Done, a brief explanation has been provided under data collection tool. A reference is added to support the statement (reference number 34). The validation was mentioned under the title validation of interview plan. The data collection tool was simple general seven questions besides demographic background and not a long questionnaire. To prepare them to collect data in a suitable way we taught the pharmacy interns who conducted the interviews how to interview a respondent in a qualitative research including asking prompting and probing questions to encourage respondent to elaborate more. The critical part was how to motivate a respondent to talk if he did not understand a question or provided noninformative answers.
Comments: There is nothing stated about the data collectors, where they trained, did they take part in the pilot study. How certain are you that they were all competent enough to collect information that is aligned with the study objectives? I also realized that the tool was not piloted on real participants but on ” a faculty staff from the college of Pharmacy, Taif University who has a long experience in a hospital setting ”. is this person a pharmacist, he/she previously engaged in the activity of GS or TI, if this is not so, why was he/she selected? For sure a faculty staff might be competent enough to advise on the process of the interview but I am not sure if they can on the type of questions to ask about GS or TI.
Authors responses: the data collectors were 5 graduating pharmacy interns who prepare the protocol of the study under the supervision of college staff. They prepared and been involved in everything related to the study and been taught on collecting data in qualitative research. They conducted the testing of the interview on the pharmacy staff. The faculty staff who helped in testing the interview was a pharmacy staff worked as a clinical pharmacist in various wards and took different positions and was a head of the pharmacy department before joining the academia. So, he has excellent experience with GS and TI. This information is added under validation of interview plan.
Comment: How was the data managed? Who did the transcribing, was it done using software or manually? Was the transcripts checked and rechecked for consistency with the recording?
Authors responses: the same 5 graduating pharmacy interns did the transcribing manually and each one checked the work of the others to confirm consistency of recoding. This is explained under thematic analysis.
The results
Comments: Line 168. There is nothing about the characteristics of the participants, here you just mentioned the number of participants from each of the hospitals.
Authors responses: the characteristics were shown in table 1. Now we added the sentence “More details on participants characteristics are shown in table 1”.
Comments: Line 176: in the results section there is no need for justifying anything only present the results. Do justification in the discussion section.
Authors responses: the statement provided was not an interpretation or a justification of finding. It is part of the findings. To make this clear we added the characteristics of the respondents who provided negative feedback regarding the presence of TI. i.e., Four of them were pharmacy technician, one clinical pharmacist with only two years of experience and one Bpharm. holder with five years’ experience. All other respondents whose jobs are clinical pharmacist, medication manager or safety officer agreed that there are both GS and TI.
Comments: Line 200 – 224, In the private setting, I did not see anything about the process of GS and TI. From the interview what was the response from pharmacists in the community to that specific question?
Authors responses: already private setting was mentioned in lines 202 (now line 254 in the edited tracked changes version) and 210 (now line 262 in the edited tracked changes version). The study did not include respondents from the community pharmacy. All are hospital pharmacy staff including those from the private sector.
Comments: line 250 “Those who said “no” are having low experience in MoH hospitals and have 250 no background of the available protocol “How did you know they have low experience in MOH protocols?
Authors responses: We removed the sentence. However, we explained that most of respondents (n=17) said yes, and few said “no” (n=4).
Discussion
Comments: There are many limitations in this study the authors did not account for there should be a sub-section in the discussion that elaborates on those limitations.
Authors responses: limitations are added now.
Conclusions
Comments: There is so much information generated yes the conclusion is redundant. Perhaps the authors might want to provide insight and recommendations for policy briefs.
Authors responses: Conclusion is modified.
Comments on the Quality of English Language
I think the English Language needs to be revised by an expert. There are some grammatical errors.
Authors responses: The English language has been revised and edited throughout the manuscript.
Submission Date
30 March 2023
Date of this review
18 Apr 2023 13:41:40
Reviewer 2 Report
Dear authors, generic subsitions is a part of pharmaceutical care. Please find more info about this. Cippole and Strand has done some job about this. Please find FIP documents on generic substition. Please think about if all medicines can be swapped to generic. What are the circumtances? Please add few insights from FIP congress there was a lot about this topic.
Author Response
Reviewer 2
Comments and Suggestions for Authors
Dear authors, generic subsitions is a part of pharmaceutical care. Please find more info about this. Cippole and Strand has done some job about this. Please find FIP documents on generic substition. Please think about if all medicines can be swapped to generic. What are the circumtances? Please add few insights from FIP congress there was a lot about this topic.
Authors responses:
Thank you very much for the supportive feedback. We searched for the document by Cipolle and Strand and the FIP and we were not able to locate them. We will be happy to include if you kindly sent them.
Swapping all medicines to generic might not be practical in all countries. Since this open the door for substitution. Some medicines require bioequivalence studies and testing to confirm equivalency among products from different manufacturers which is not available facility in all countries. However, this is beyond the scope of our study. The scope of our study was simply seeing how generic substitution and therapeutic interchange are practiced and whether there are supportive policies or not. Thank you very much.
Submission Date
30 March 2023
Date of this review
15 May 2023 09:34:22
Reviewer 3 Report
1- please make abstract structured. background, method, results, conclusion
2- the introduction is massive. Please try to summarize and combine paragraphs if you can
3- please add paragraph on the limitation of the study
1- please make abstract structured. background, method, results, conclusion
2- the introduction is massive. Please try to summarize and combine paragraphs if you can
3- please add paragraph on the limitation of the study
Author Response
Reviewer 3
Comments on the Quality of English Language
Authors responses: The authors would like to thank the reviewer for his constructive comments and supportive feedback that have contributed much to the improvement of the manuscript.
1- please make abstract structured. background, method, results, conclusion
Authors responses: done as suggested
2- the introduction is massive. Please try to summarize and combine paragraphs if you can
Authors responses: done. We removed the unnecessary paragraphs from the introduction and tried to improve the quality of the English Language.
3- please add paragraph on the limitation of the study
Authors responses: done, limitations were added.
Submission Date
30 March 2023
Date of this review
22 May 2023 21:47:47
Reviewer 4 Report
Thank you very much for the opportunity to review this article about generic substitutions and therapeutic interchanges in hospital pharmacies. Overall I found it to be a bold article that offers a different approach and, therefore, may offer a valuable perspective for clinical practice.
However, some minor aspects are amenable to revision. These are as follows:
In the Methods section, I propose that at the beginning, they describe in a general way the type of study being performed. Then, the eligible population and its characteristics. The latter, together with the selection criteria, is important to be well described since one of the main problems of this type of study lies precisely in the potential for selection bias.
In fact, in 2.1 Setting and Study Design (page 3 of 12, lines 119-122): When was the study done?
Or, 2.3 Sampling (page 3 of 12, lines 134-135): Which technique was used? "Hospital pharmacists were selected using a convenient sampling technique within a group of hospitals from each of these selected cities."
Although specific characteristics of the sample, such as size, are discussed in the limitations, evaluating the potential selection bias would be important.
Author Response
Reviewer 4
Comments and Suggestions for Authors
Thank you very much for the opportunity to review this article about generic substitutions and therapeutic interchanges in hospital pharmacies. Overall I found it to be a bold article that offers a different approach and, therefore, may offer a valuable perspective for clinical practice.
However, some minor aspects are amenable to revision. These are as follows:
Authors responses: The authors would like to thank the reviewer for his constructive comments and supportive feedback that have contributed much to the improvement of the manuscript
In the Methods section, I propose that at the beginning, they describe in a general way the type of study being performed. Then, the eligible population and its characteristics. The latter, together with the selection criteria, is important to be well described since one of the main problems of this type of study lies precisely in the potential for selection bias.
Authors responses: More elaboration was made in the methods section under various subheadings. The study involved pharmacy staff from different backgrounds regarding the nature of jobs, the type of hospital and experiences. We targeted the main hospitals in the region where our pharmacy student receive training regularly.
In fact, in 2.1 Setting and Study Design (page 3 of 12, lines 119-122): When was the study done?
Authors responses: We added the time frame of the data collection.
Or, 2.3 Sampling (page 3 of 12, lines 134-135): Which technique was used? "Hospital pharmacists were selected using a convenient sampling technique within a group of hospitals from each of these selected cities."
Authors responses: We targeted the main hospitals in the region where our pharmacy student receive training regularly. Within hospitals all pharmacists found available at times of interviewers’ visits were taken. This was how convenience sampling technique was undertaken. More description was added under title “sampling”.
Although specific characteristics of the sample, such as size, are discussed in the limitations, evaluating the potential selection bias would be important.
Authors responses: We added a limitation section. Sample size was not a limitation of this study. In qualitative research recruitment is stopped when saturation is achieved. This is described also under the subtitle “Sample Size” and we provided a reference.
Submission Date
30 March 2023
Date of this review
23 May 2023 17:14:52